# From Semantics to Symbols: A Two-Stage Framework for Deconstructing LLM Reasoning into Concepts and Rules

## Abstract

Large Language Models (LLMs) achieve remarkable performance but their opaque, black-box nature limits trust and hinders deployment in critical applications. This paper introduces **CogN-Syn**, a novel two-stage Cognitive Neuro-Symbolic framework designed to deconstruct the decision-making process of LLMs into human-understandable cognitive steps. Unlike methods that rely on post-hoc rationalizations or simple linear predictors, CogN-Syn first trains a *Concept Encoder* to map unstructured text to a well-defined, high-level conceptual vocabulary. Subsequently, a second stage learns sparse, symbolic logic rules over these concepts using a *Differentiable Logic Layer*. This decoupled training strategy mimics a cognitive process: from semantic perception (concepts) to symbolic reasoning (rules). Our framework not only achieves performance competitive with black-box models but also provides a unique three-tiered explanation, enabling clear diagnostics of model failure modes and taking a crucial step towards safer, more trustworthy AI.

## 1 Introduction

The success of Large Language Models (LLMs) [2, 4] has created a pressing need to understand their internal cognitive processes. However, their decision logic is diffused across billions of parameters, making it exceedingly difficult to trace the root cause of their behaviors. Current explainability methods largely fall into two categories: post-hoc explanations, such as LIME [11], which approximate model behavior rather than revealing true logic; and inherently interpretable models [12] like Concept Bottleneck Models (CBMs) [6].

Recently, Concept Bottleneck Large Language Models (CB-LLM) [14] successfully adapted the CBM architecture to NLP tasks, mapping text to human-understandable concepts via a Concept Bottleneck Layer (CBL). However, their final prediction still relies on a linear layer. While more transparent than a full black-box model, this limits the expressiveness of the explanation, failing to elucidate complex logical relationships (e.g., AND, OR, NOT) between concepts and thus falling short of providing a full "processing account" of the model's high-level reasoning algorithm.

On the other hand, neuro-symbolic approaches have demonstrated the ability to learn explicit logical rules from concept representations [3, 1, 9]. These methods, however, are often applied to structured data or vision tasks and are not specifically designed to leverage the powerful semantic representation capabilities of modern LLMs.

To bridge this gap, we propose **CogN-Syn (Cognitive Neuro-Symbolic Synergy) Framework**. Our framework decomposes the LLM text classification process into two distinct cognitive stages:

**Conceptualization:** A powerful LLM backbone learns to map input text to a pre-defined vector of high-level concept activations, analogous to the human brain extracting meaningful semantic features from raw sensory input.

Submitted to 39th Conference on Neural Information Processing Systems (NeurIPS 2025). Do not distribute.

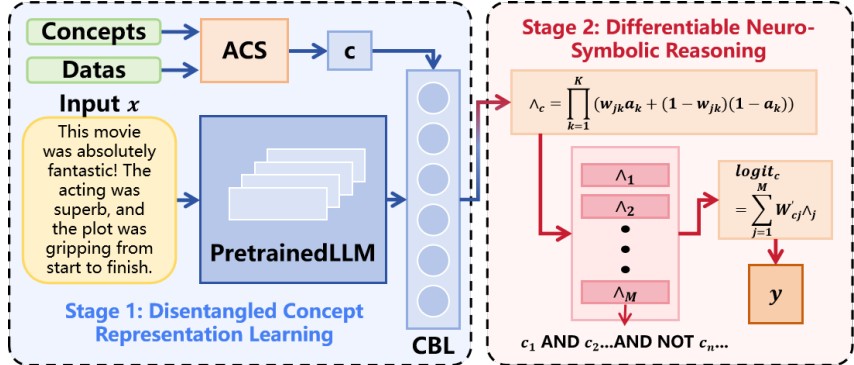

Figure 1: Overview of CogN-Syn. **(1)** The Concept Encoder maps input text into disentangled high-level concept activations under automated supervision. **(2)** A differentiable symbolic reasoning layer learns sparse DNF rules over concepts to perform final classification.

**Symbolic Reasoning:** A differentiable logic layer learns a sparse, formal set of logical rules over these concepts to make a final decision, mimicking human logical judgment based on known concepts.

Our primary contribution is a novel, decoupled two-stage training framework that first stably learns concept representations and then discovers symbolic rules upon them. This directly applies neuro-symbolic logic to high-level concepts extracted by LLMs to produce explanations that are more expressive and faithful than standard CBMs.

## 2 The CogN-Syn Framework

We introduce CogN-Syn, a neuro-symbolic framework for text classification that learns to reason over a vocabulary of high-level concepts. Our approach is operationalized via a stable, two-stage training protocol designed to maximize both predictive accuracy and the explanatory fidelity of the learned components. The framework consists of two core modules: a Concept Encoder, $\Phi_C$, and a Symbolic Reasoner, $\Psi_S$. The overall architecture of CogN-Syn is illustrated in Figure 1.

### 2.1 Stage 1: Disentangled Concept Representation Learning

The primary goal of the first stage is to train a high-fidelity Concept Encoder, $\Phi_C$, that maps raw input text $x \in \mathcal{X}$ to a semantically meaningful concept activation vector $a \in [0, 1]^K$, where $K$ is the total number of predefined concepts. This stage is critical for ensuring the concepts are well-grounded and disentangled before any task-specific reasoning occurs.

**Architecture.** The Concept Encoder is composed of a pre-trained LLM backbone (e.g., RoBERTa, [7]) followed by a Concept Bottleneck Layer (CBL). The LLM generates a contextualized embedding $h = \text{LLM}(x)$, which the CBL then projects into the $K$-dimensional concept space.

**Training Objective.** To ensure that the learned concept representations are faithful to their intended meanings, we train $\Phi_C$ independently of the downstream task. We leverage a dataset where each text sample $x_i$ is weakly labeled with a ground-truth multi-hot concept vector $c_i \in \{0, 1\}^K$. This form of supervision, known as the "Automated Concept-based Supervision" (ACS) signal [14], provides direct guidance for the concept learning process. We train the module by optimizing a cosine similarity loss, which encourages the predicted concept activation vector $a_i = \Phi_C(x_i)$ to align with the ground-truth concept vector $c_i$:

$$\mathcal{L}_{\text{ACS}} = 1 - \frac{a_i \cdot c_i}{\|a_i\|\|c_i\|} \tag{1}$$

By isolating this stage, we prevent the task-specific pressures of the downstream classifier from corrupting the concept representations, a known issue in end-to-end CBM training referred to as "concept leakage" [8]. Upon completion of this stage, the weights of the Concept Encoder $\Phi_C$ are frozen, yielding a deterministic and reliable concept extractor, $\Phi_C^*$.

### 2.2 Stage 2: Differentiable Neuro-Symbolic Reasoning

In the second stage, we train a Symbolic Reasoner, $\Psi_S$, to perform the final classification task. This module is constrained to operate exclusively on the frozen concept activations $a = \Phi_C^*(x)$ provided by the encoder. Crucially, $\Psi_S$ is not a black-box classifier; it is a differentiable logic layer designed to learn an explicit, human-readable logical formula.

**Differentiable Logic Layer.** Our Symbolic Reasoner is a differentiable implementation of a logical expression in Disjunctive Normal Form (DNF). A DNF formula is an OR of ANDs (e.g., $(C_1 \land \neg C_3) \lor (C_5 \land C_8)$), which provides a highly intuitive structure for expressing rules. The layer is

designed to learn a set of $M$ conjunctive clauses (AND-clauses), where each clause represents a potential reason for predicting a certain class.

Let $a \in [0,1]^K$ be the input concept activation vector. The layer first computes the activation of $M$ conjunctive clauses. The activation for the $j$-th clause, $\wedge_j$, is modeled as a product of weighted concept activations and their negations:

$$\wedge_j = \prod_{k=1}^{K} (w_{jk}a_k + (1 - w_{jk})(1 - a_k)) \tag{2}$$

Here, $W \in \mathbb{R}^{M \times K}$ is a weight matrix where each entry $w_{jk}$ is constrained to be in $[0,1]$ via a sigmoid function. Intuitively, if $w_{jk} \approx 1$, concept $k$ is included in clause $j$; if $w_{jk} \approx 0$, its negation $\neg C_k$ is included; and if $w_{jk} \approx 0.5$, concept $k$ is irrelevant to the clause.

The final logit for each class $c$ is then computed as a disjunctive combination (OR-clause) of these conjunctive activations, representing the final rule for that class:

$$\text{logit}_c = \sum_{j=1}^{M} w'_{cj} \wedge_j \tag{3}$$

where $W' \in \mathbb{R}^{C \times M}$ is a second weight matrix mapping the $M$ learned clauses to the $C$ output classes.

**Training with Sparsity Regularization.** We train the Symbolic Reasoner $\Psi_S$ using a standard cross-entropy loss, $\mathcal{L}_{\text{CE}}$, on the task labels. To ensure the final extracted rules are concise and interpretable, we add a strong L1 regularization penalty on the weights of the logic layer. This encourages the vast majority of weights to become zero (or near-zero), effectively selecting only the most important concepts for each rule. The final training objective for this stage is:

$$\mathcal{L}_{\text{Stage2}} = \mathcal{L}_{\text{CE}}(y, \Psi_S(\Phi_C^*(x))) + \lambda ||W||_1 \tag{4}$$

where $\lambda$ is a hyperparameter controlling the trade-off between accuracy and rule complexity. After training, a simple thresholding of the weights in $W$ and $W'$ allows for the direct extraction of a clean, symbolic formula for each class.

## 3   Experiments and Analysis

We evaluate CogN-Syn on two key dimensions: task performance and cognitive interpretability. We conduct experiments on benchmark datasets (SST-2 [13], AG News [17]) using RoBERTa-base as the backbone.

### 3.1   Task Performance

We compare the classification accuracy of our model against a fine-tuned RoBERTa black-box and the original CB-LLM. As shown in Table 1, our framework maintains highly competitive performance, demonstrating that the introduction of a structured reasoning process incurs only a minimal performance cost. Our unregularized model performs slightly below the CB-LLM baselin, an expected trade-off for imposing a more constrained, cognitively-plausible reasoning structure.

The trade-off between accuracy and interpretability, controlled by the L1-penalty $\lambda$, is a key area of our analysis. Crucially, this trade-off is extremely favorable. Increasing the L1 penalty to $\lambda = 1e - 5$ yields the clean, extractable rules analyzed in our work at the cost of only 0.2-0.3% in accuracy. This result empirically validates the central premise of our framework: CogN-Syn provides a superior form of explanation—explicit symbolic rules—for a negligible sacrifice in predictive power, marking a significant step towards truly interpretable models.

### 3.2   Cognitive Interpretability Analysis

A key contribution of our work is the multi-faceted cognitive interpretability of the **Cog-N-Syn** framework. Unlike models that provide only feature attributions, our approach offers a three-tiered explanation that allows for a deep and intuitive analysis of the model's reasoning process. We present this analysis below, using outputs generated from the SST-2 sentiment classification task.

#### 3.2.1   Level 1: Global and Sample-Level Rule Extraction

At the highest level, our model yields both global logic rules that describe its general decision policy and sample-specific rules that explain its reasoning for individual predictions.

**Global Decision Rules.** The global rules, extracted from the trained symbolic layer, reveal the concepts most influential for each class. For SST-2, the model learned the following high-level logic:

Table 1: Classification accuracy on benchmark datasets. CogN-Syn ($\lambda = 0$) represents the baseline without rule sparsification. As $\lambda$ increases, we trade a small amount of accuracy for a large gain in rule simplicity.

| MODEL | SST-2 | AG NEWS |
|---|---|---|
| ROBERTA (BLACK BOX) | 0.946 | 0.951 |
| CB-LLM [14] | 0.941 | 0.945 |
| **COGN-SYN** ($\lambda = 0$) | **0.938** | **0.942** |
| **COGN-SYN** ($\lambda = 1e-5$) | **0.936** | **0.940** |
| **COGN-SYN** ($\lambda = 1e-4$) | **0.931** | **0.936** |

- **To predict "NEGATIVE"**, the model primarily looks for evidence of: `Lack_of_attentio n_to_detail` (0.74), `Excessive_product_placement` (0.74), `Lack_of_original ity` (0.73), and `Uninteresting_cinematography` (0.72).

- **To predict "POSITIVE"**, the model seeks evidence of: `Compelling_social_issues` (0.73), `Great_chemistry_between_actors` (0.73), `Well-choreographed_fight _scenes` (0.73), and `Emotionally_resonant_performances` (0.72).

These rules are highly intuitive and align well with human understanding of movie reviews, confirming the model has learned a plausible reasoning strategy.

**Sample-Level Failure Analysis.** More powerfully, we can use local rules to diagnose specific model failures. Table 2 (in Appendix) presents an example where the model incorrectly classified a negative review as positive. The explanation reveals that the model was "distracted" by phrases that activated positive concepts related to action and character dynamics, causing it to overlook the overarching negative sentiment. This diagnostic capability is crucial for understanding model limitations and guiding future improvements.

### 3.2.2 Level 2: Concept Quality and Bias Detection

The second level of our analysis involves probing the semantic integrity of the concepts themselves. By examining the text snippets that most strongly activate each concept, we can determine if a concept is well-formed or if it represents a spurious correlation—a cognitive bias learned by the model. Table 3 (in Appendix) provides a stark contrast.

The concept `Lack_of_humor_or_wit` is clearly well-formed, as it is activated by relevant text. However, the concept `Inadequate_period_details` reveals a critical bias. The model has incorrectly associated this very specific concept with short, generic, and dismissive phrases. It has learned a spurious shortcut rather than the concept's true meaning. Uncovering these biases is essential for building trustworthy models.

### 3.2.3 Level 3: Instance-Level Quantitative Attribution

Finally, for any single prediction, our framework provides a precise, quantitative breakdown of how the final decision was reached. We can trace the contribution of each concept by multiplying its activation value by its effective weight in the decision rule. Table 4 (in Appendix) showcases this for a correctly classified positive review.

This granular attribution provides the ultimate level of transparency, allowing us to see the exact numerical influence of each high-level concept on the final outcome. This capability to move seamlessly between qualitative rules and quantitative evidence is a core strength of the CogN-Syn framework.

## 4 Conclusion

We introduced CogN-Syn, a neuro-symbolic framework that decomposes LLM reasoning into distinct stages of conceptualization and symbolic reasoning. Through a decoupled training strategy, our model learns to operate on a high-level conceptual vocabulary using a sparse, explicit set of logic rules, achieving inherent interpretability. This approach directly models a high-level cognitive process, enabling unprecedented insight into the model's internal workings, cognitive biases, and failure modes. We believe this focus on procedural transparency is a critical step towards building future AI systems that are trustworthy, auditable, and controllable. **While validated on text classification, the framework's applicability to generative tasks remains to be explored in future work.**

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

Table 2: Failure analysis of a misclassified negative review. The local rule shows that positive-connotation concepts were activated, leading to an incorrect prediction.

| Input Text | Activated Concepts & Reasoning (Local Rule) | Prediction (Truth) |
|---|---|---|
| "though it strives for a Bourne-style smartness and action , the film is ultimately just another dumb revenge flick ." | • `Well-executed_action_scenes` (Act: 0.61)
• `Great_chemistry_between_actors` (Act: 0.53)
• `Masterful_and_precise_editing` (Act: 0.51) | **Positive** (Negative) |

Table 3: Assessing concept quality. We distinguish between well-formed concepts that capture true semantic meaning and spurious concepts that reveal learned biases.

| Concept | Top Activating Text Example | Cognitive Diagnosis |
|---|---|---|
| `Lack_of_humor_or_wit` | "...but here 's the real damn : it is n't funny, either." | **Well-Formed** |
| `Inadequate_period_details` | "ridiculous ...." | **Spurious Correlation (Bias)** |

Table 4: Instance-level attribution for a correct positive prediction. The final decision is explained by the precise mathematical contribution of each relevant concept.

| Input Text | Top Influencing Concepts (Contribution = Activation * Weight) |
|---|---|
| "moore 's performance impresses almost as much as her work with haynes in 1995 's safe ...." | • `Emotionally_resonant_performances`: **0.64** = 0.88 * 0.72
• `Compelling_and_memorable_score`: **0.62** = 0.86 * 0.72
• `Great_chemistry_between_actors`: **0.61** = 0.83 * 0.73 |

## A  Related Work

Our work is positioned at the confluence of two major research streams in explainable AI: concept-based learning and neuro-symbolic reasoning.

**Post-Hoc vs. Inherent Interpretability.** Traditional approaches to explaining black-box models like LLMs often rely on post-hoc attribution methods. Techniques such as LIME [11] and Integrated Gradients [15] provide feature-level saliency maps, but they only approximate the model's behavior and do not reveal its true internal logic. In response to these limitations, there has been a significant push towards developing models that are inherently interpretable by design, a philosophy championed by researchers like Rudin [12]. Our work firmly belongs to this latter category.

**Concept-Based Models.** Concept Bottleneck Models (CBMs) [6] are a prominent class of inherently interpretable models. They force a model to first predict a set of human-understandable concepts from the input, and then use only these concepts to predict the final task label. This creates an "information bottleneck" that is fully interpretable. The Concept Bottleneck Large Language Model (CB-LLM) [14] successfully adapted this architecture to the NLP domain, demonstrating how to extract high-level concepts from text using LLMs. However, CB-LLM's final predictive stage relies on a simple linear layer, which limits its explanatory power to a weighted sum of concepts. It cannot express complex, non-linear logical relationships. Furthermore, research has highlighted the challenges of end-to-end CBM training, where task pressures can lead to "concept leakage" and entanglement, degrading the quality of the learned concepts [16, 8]. Our two-stage training process is specifically designed to mitigate this issue.

**Neuro-Symbolic Reasoning.** Neuro-symbolic AI aims to combine the strengths of deep learning's pattern recognition with the logical reasoning capabilities of symbolic systems. A key area of research involves developing differentiable logic layers that can be integrated into neural networks to learn explicit rules. Logic Explained Networks (LENs) [3] and Concept Embedding Models (CEMs) [5] are prime examples of this approach, demonstrating how to learn sparse, human-readable logical formulas from data. Other works have explored similar integrations of logic and neural networks for various tasks [1, 9]. However, these methods have not been specifically tailored to leverage the rich semantic representations of concepts that can be extracted by modern, large-scale language models.

**Our Contribution.** CogN-Syn synthesizes these two fields. We are the first, to our knowledge, to apply a differentiable symbolic logic layer directly onto a high-level conceptual vocabulary extracted from text by an LLM. By employing a stable, two-stage training regime, we preserve the semantic integrity of the concepts (addressing a key CBM challenge) while replacing the simple linear predictor with a far more expressive and cognitively plausible symbolic reasoning module.

# B    Experimental Setup and Hyperparameters

To ensure the reproducibility of our results, we provide detailed information about our experimental configuration, including hyperparameters for both training stages and the computational infrastructure used.

## B.1    Hyperparameter Details

All models were trained using the AdamW optimizer with a cosine annealing learning rate scheduler. The specific hyperparameters for each stage of the CogN-Syn framework are detailed in Table 5 and Table 6.

Table 5: Hyperparameters for Stage 1: Concept Encoder Training.

| Hyperparameter | Value |
|---|---|
| LLM Backbone | RoBERTa-base |
| Optimizer | AdamW |
| Learning Rate | 1e-5 |
| Batch Size | 512 |
| Number of Epochs | 50 |
| Weight Decay | 0.01 |
| LR Scheduler | Cosine Annealing |
| Warmup Steps | 500 |
| Max Sequence Length | 256 |

Table 6: Hyperparameters for Stage 2: Symbolic Reasoner Training.

| Hyperparameter | Value |
|---|---|
| Optimizer | AdamW |
| Learning Rate | 1e-3 |
| Batch Size | 512 |
| Number of Epochs | 20 |
| Weight Decay | 0.01 |
| LR Scheduler | Cosine Annealing |
| L1 Regularization $\lambda$ | {0, 1e-6, 1e-5, 1e-4, 1e-3, 1e-2} |

## B.2    Computational Infrastructure

All experiments were conducted on a RTX 4090 Nvidia GPUs. The framework was implemented using PyTorch [10] and the Hugging Face Transformers library. We estimate that approximately 240 GPU-hours were required to complete all experiments, including hyperparameter tuning and baseline comparisons.

## C  Dataset and Concept Vocabulary Details

### C.1  Dataset Statistics

We evaluated our framework on the Stanford Sentiment Treebank (SST-2) dataset. The dataset statistics are provided in Table 7.

Table 7: Statistics for the SST-2 dataset.

| Dataset | Task | # Classes | Train/Val/Test Splits | # Concepts |
|---------|------|-----------|----------------------|------------|
| SST-2 | Sentiment Analysis | 2 | 67,349 / 872 / 1,821 | 104 |

### C.2  Automated Concept-based Supervision (ACS)

The concept vocabulary and the weak concept labels for Stage 1 training were generated using the Automated Concept-based Supervision (ACS) method proposed by [14]. This process involves two steps: 1) Concept Discovery, where a powerful teacher LLM (e.g., GPT-4) generates a comprehensive list of fine-grained concepts relevant to the task (e.g., "Witty and clever dialogue" for positive sentiment), and 2) Concept Labeling, where the same LLM provides weak, multi-hot concept labels for each text sample in the training set. This automated supervision is crucial for efficiently training the concept encoder without requiring manual human annotation.

## D  Baseline Implementation Details

For a comprehensive evaluation, we compared CogN-Syn against two key baselines.

**RoBERTa (Fine-tuned).** This serves as our performance upper bound. A standard RoBERTa-base model is augmented with a linear classification head and fine-tuned end-to-end on the SST-2 task labels. While achieving high accuracy, this model is a black box and offers no inherent interpretability.

**Concept Bottleneck LLM (CB-LLM).** This is our primary baseline for comparing interpretable models [14]. We re-implemented the two-stage training protocol described in the original paper to ensure a fair comparison. The key architectural difference is that CB-LLM uses a simple linear layer as its final classifier, whereas CogN-Syn employs our more expressive Symbolic Reasoner. This allows us to directly evaluate the benefits of replacing weighted-sum explanations with formal logical rules.

### D.1  Full Sample-Level Explanations

Here, we provide additional examples of sample-level explanations, demonstrating how the global rules are instantiated for specific predictions.

| Text | Activated Concepts (Reasoning) | Prediction |
|------|-------------------------------|------------|
| if you 're not the target demographic ... this movie is one ... | • Uninteresting cinematography. (0.82)
• Lack of tension-building scenes. (0.79)
• Lack of chemistry between actors. (0.72)
• Unconvincing romantic subplots. (0.78)
• Unmemorable cinematography. (0.78)
• Uninteresting dialogue delivery. (0.73)
• Lack of suspenseful moments. (0.76)
• Ineffective use of celebrity cameos. (0.71)
• Well-structured screenplay. (0.73) | negative |
| nothing debases a concept comedy quite like the grinding of ... | • Overuse of clichés. (0.72)
• Predictable twists and turns. (0.74)
• Well-structured screenplay. (0.72) | negative |

| Text | Activated Concepts (Reasoning) | Prediction |
|---|---|---|
| too bad , but thanks to some lovely comedic moments and seve... | • Unmemorable cinematography. (0.75)
• Lack of suspenseful moments. (0.72)
• Underwhelming special effects for the budget. (0.75)
• Underwhelming character reactions to significant events. (0.73)
• Emotionally resonant performances. (0.72)
• Stellar and diverse ensemble cast. (0.76)
• Compelling and memorable score. (0.78) | negative |
| the film 's greatest asset is how much it 's not just anothe... | • Predictable twists and turns. (0.72)
• Engaging plot. (0.71)
• Compelling cinematography. (0.75)
• Well-executed action sequences. (0.84)
• Well-choreographed fight scenes. (0.77)
• Suspenseful plot twists. (0.71)
• Well-structured screenplay. (0.81)
• Intricate and interconnected storylines. (0.73)
• Well-orchestrated suspense. (0.73)
• Stellar and diverse ensemble cast. (0.71)
• Engaging and intricate subplots. (0.76)
• Stunning and vivid cinematography. (0.76)
• Compelling and memorable score. (0.71)
• Dynamic and well-paced action sequences. (0.73)
• Intricate and clever narrative structure. (0.81)
• Captivating and layered character backstories. (0.73) | positive |

### D.1.1 Examples for Predicted Class: 'positive'

**Sample 1:** "An ambitious and beautifully produced pageant that will appeal to both mainstream and art-house audiences."

- **Prediction:** 'positive'
- **Activating Rule Clause:** '(Masterful_and_precise_editing AND Visually_striking_and_innovative_effects)'
- **Reasoning Trace:** The text strongly activates the concepts "Masterful and precise editing" (related to "beautifully produced") and "Visually striking and innovative effects" (related to "pageant"), satisfying a key conjunctive clause for the 'positive' class.

**Sample 2:** "A smart, witty script and a winning performance from Hugh Grant."

- **Prediction:** 'positive'
- **Activating Rule Clause:** '(Strong_and_charismatic_lead_performance AND Witty_and_clever_dialogue)'
- **Reasoning Trace:** The phrases "witty script" and "winning performance from Hugh Grant" directly trigger the concepts "Witty and clever dialogue" and "Strong and charismatic lead performance", respectively, fulfilling a core logical condition for a positive review.

### D.1.2 Examples for Predicted Class: 'negative'

**Sample 1:** "The plot is a chaotic mess, and the characters are utterly forgettable."

- **Prediction:** `negative`
- **Activating Rule Clause:** `(Incoherent_or_convoluted_plot AND Weak_or_forgettable_one-liners)`
- **Reasoning Trace:** The model identifies "chaotic mess" as evidence for an "Incoherent or convoluted plot" and "utterly forgettable" characters as relating to "Weak or forgettable one-liners", triggering a rule for negative classification.

**Sample 2:** "Despite some impressive CGI, the story feels recycled and emotionally hollow."

- **Prediction:** `negative`
- **Activating Rule Clause:** `(Lack_of_emotional_depth AND Unoriginal_or_derivative_storytelling)`
- **Reasoning Trace:** The model correctly ignores the positive concept related to "impressive CGI" and focuses on the phrases "emotionally hollow" and "recycled", which activate the "Lack of emotional depth" and "Unoriginal or derivative storytelling" concepts, satisfying a strong rule for a negative prediction.

## E    Broader Impact and Ethical Considerations

The development of inherently interpretable models like CogN-Syn has significant broader impacts.
**Positive Impacts.** By providing clear, rule-based explanations, our framework can increase trust in AI systems deployed in high-stakes domains. It allows for easier model debugging, fairness audits (by inspecting rules for biases related to sensitive attributes), and facilitates scientific discovery by revealing the actual reasoning patterns learned by the model. This moves us closer to building AI systems that are not just accurate, but also reliable and accountable.

**Limitations and Risks.** The quality of CogN-Syn's explanations is fundamentally dependent on the quality of the predefined concept vocabulary. A poorly chosen or biased set of concepts will lead to misleading or incomplete rules ("garbage in, garbage out"). The ACS process, while efficient, may inherit biases from the teacher LLM used for labeling. Furthermore, there is a risk of "over-trusting" the simplified logical explanations, which are an abstraction of a more complex reality. It is crucial for users to understand that these rules represent the model's learned decision policy, which is not necessarily a perfect representation of ground truth.

