# OpenReview forum: "From Semantics to Symbols: A Two-Stage Framework for Deconstructing LLM Reasoning into Concepts and Rules"
_NeurIPS.cc/2025/Workshop/Reliable_ML — NeurIPS 2025 - Reliable ML Workshop_

### Official Review · Reviewer_3Nk3 · 2025-09-16
**Interesting, internally consistent system with some flaws preventing this being a more substantial leap forward in understandability**

**Rating:** 6
**Confidence:** 2

**Review:**

## Paper Summary

The authors introduce CogN-Syn, a framework for improving the explainability of LLM decision making. Specifically, they use a two-stage approach. The first is "conceptualization" which converts raw text into high level concepts. The second is "symbolic reasoning" which applies logic against the concepts derived for some text in stage 1 to make certain predictions. These can be mapped to consistent "rules" that explain the decision making process of the LLM for some input. Crucially, CogN-Syn allows breaking down the decision making for a single input against the concepts and their logical relationships.

## Strengths

The paper is fairly well written. It seems like it's a good fit for the venue given the importance of being able to understand how a model arrived at a particular decision to safety and alignment. The paper seems rigorous (within the bounds set by the authors themselves).

## Limitations

The authors call this out in Appendix E but I think more emphasis is merited; CogN-Syn's value is strictly dependent on the concepts extracted in stage 1, which itself relies on another LLM. At the very least, I think this needs to be in the main body of the paper with a more complete treatment. I'm curious if the authors could run an experiment with an obviously biased concept vocabulary to understand the resilience of CogN-Syn as a whole.

I also suggest expanding on 3.2.2 to highlight that detection of biased concepts is not intrinsic to CogN-Syn (unless I misunderstood?). Rather, it relies on a human interpreting the activated concepts to determine that a particular concept is biased.

## Suggestions to Authors

* Consider greater emphasis on the significance of concept vocabulary training in the main paper
* Consider experimenting with different qualities of concept vocabulary to understand tangible impact on CogN-Syn
* Expand 3.2.2 to discuss the bounds of "detecting" bias in concepts
* A discussion of how the approach maps to NAP (https://arxiv.org/pdf/2206.10611) despite the domains differing

Moonshot: The paper might benefit from a brief discussion of how CogN-Syn can be used recursively and repeatedly in stage 1 runs. Is it possible to arrive at a substantially smaller "seed" LLM from which we can repeatedly derive interpretable LLMs (analogous to bootstrappable builds in the world of compilers)? I understand this may be a significant lift for this paper and venue, but a discussion of whether this is possible, can be explored in future work, etc. may be helpful.

## Ethics

Nothing come to mind other than what I've mentioned regarding emphasizing the importance of the concept vocabulary which itself is created via a less understandable LLM.

---

### Official Review · Reviewer_Zdnn · 2025-09-20
**Review for "back arrowBack to Reviewers Console From Semantics to Symbols: A Two-Stage Framework for Deconstructing LLM Reasoning into Concepts and Rules"**

**Rating:** 6
**Confidence:** 2

**Review:**

The paper studies the problem of interpretability in large language models. Specifically, for NLP applications, the Concept Bottleneck architecture was recently adapted to map text to human-understandable concepts via a Concept Bottleneck layer, but this architecture still relies on a linear layer for their final prediction. The authors posit that this fails to explicate logical relationships between concepts and thus the model’s high-level reasoning algorithm is not fully interpretable.

The paper introduces CogN-Syn, a neuro-symbolic framework that decomposes LLM reasoning into two stages: concept learning and symbolic rule extraction. The problem is well motivated, as black-box behavior limits trust in LLMs, especially in high-stakes domains. The authors’ approac of first mapping text into a disentangled concept space and then applying a differentiable logic layer to learn explicit, human-readable rules offers a structured way to align model reasoning with cognitive processes. More importantly, the results are definitely promising, as CogN-Syn achieves accuracy comparable to strong baselines while providing interpretable, rule-based explanations.

Strengths:

** The two-stage training mitigates common pitfalls of concept bottleneck models, such as concept leakage, while still leveraging the semantic richness of LLMs.

** The symbolic reasoning component produces explanations that are both global and local, allowing for diagnostics of both broad decision patterns and individual failures.

Weaknesses

** While demonstrated on classification tasks, the applicability of CogN-Syn to more complex generative tasks remains untested, which narrows its current scope relative to the broader interpretability literature.

Overall, this is a well-motivated and interesting idea that serves as a starting ground for more meaningful contributions towards explainable LLMs. The framework seems thoughtfully designed, the trade-offs between interpretability and accuracy appear minor, and the paper is rigorous and insightful.